# Building Information Modeling in Quebec's Procurement for Public Infrastructure: A Case for Integrated Project Delivery

**Gabriel Jobidon [1,2,\*], Pierre Lemieux [2] and Robert Beauregard [3]**

1   CIRCERB–CRMR, Université Laval, 2325 Rue de l'Université, Ville de Québec, QC G1V 0A6, Canada
2   Faculty of Law, Université Laval, 2325 Rue de l'Université, Ville de Québec, QC G1V 0A6, Canada; pierre.lemieux@fd.ulaval.ca
3   Academic and Student Affairs, Université Laval, 2320 Rue des Bibliothèques, Ville de Québec, QC G1V 0A6, Canada; vice-recteur@vre.ulaval.ca
\*   Correspondence: gabriel.jobidon.1@ulaval.ca; Tel.: +1-581-992-6478

**Abstract:** The Province of Quebec is currently in the process of adopting building information modeling (BIM) for major infrastructure projects. However, legal and contractual concerns such as the tendering process, adjudication criteria, intellectual property and risk–reward sharing mechanisms hinder the implementation of an efficient BIM process. This paper addresses the following question: How do norms, whether legislative, regulatory or contractual, functionally or dysfunctionally affect the effective implementation of BIM in Quebec's public infrastructure framework? This paper suggests that the use of Integrated Project Delivery (IPD) should help mitigate legal barriers hindering BIM implementation, while preserving balance between fairness and encouraging collaboration. Quebec's normative framework, which includes legislation, regulations, contracts and infra-regulatory rules, should be modified to standardize collaborative mechanisms, integrate two-stage negotiated processes such as rank-and-run or best and final offer and enable the assessment of tenderers' objective qualities and more subjective qualities. Furthermore, a risk–reward sharing mechanism should be implemented through target costing, and upstream participation from a wide range of stakeholders should be encouraged.

**Keywords:** building information modeling; integrated project delivery; public procurement; collaboration; infrastructure contracts

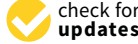



## 1. Introduction

In the past two decades, the productivity of the Canadian manufacturing industry has nearly doubled, whereas in construction it remained stagnant (McKinsey & Company 2017). To help achieve better productivity, the construction industry has turned towards building information modeling (BIM) (Succar 2009). BIM is a digital technology to establish a computable representation of all the physical and functional characteristics of a facility and its related project/life-cycle information, intended to be a repository of information for the facility owner/operator to use and maintain throughout the life-cycle of the facility (NBIMS 2007). The BIM process is essentially a method to align design members of a construction project and ensure their collaboration through information-sharing, notably through a multi-dimensional 3D model providing visual and physical properties of the asset, which can be used throughout the life-cycle of the infrastructure (Attrill and Mickovski 2020).

BIM maturity levels are defined within a range from 0 to 3. Level 0 means no collaboration and the use of traditional 2D drafting, while level 1 implies low collaboration between different stakeholders who are individually responsible for creating and managing their own data. Level 2 promotes collaborative working by ensuring each party is responsible for a 3D model which will then be combined in a federated BIM Model. Level 3 BIM involves multidisciplinary work and needs contractual frameworks encouraging open

and collaborative working and the creation of a cooperative environment throughout the life-cycle of projects. This third level enables all stakeholders to work simultaneously on the same model, therefore greatly diminishing the chance of conflicting information (Sacks et al. 2018). Widespread adoption of BIM and Integrated Project Delivery (IPD) in the public sector has notably been hindered by legal barriers (Ghassemi and Becerik-Gerber 2011). Many studies have pointed out that in order to effectively implement BIM, legal barriers need be overcome, such as liability and risk allocation issues and the status of intellectual property of the model, as well as the inadequacy of procurement practices and contracts (Sacks et al. 2018). Moreover, recent studies found that significant legal aspects or contract provisions need to be included in BIM contracts (Chihib et al. 2019), and that design–bid–build procurement and a lack of standardization impedes effective adoption of BIM (Fan et al. 2018; Leśniak et al. 2021). These issues are all addressed in the present paper, notably through the lens of IPD. IPD is defined as a contractually based approach, which creates an environment that enhances collaboration, innovation and value, and which is characterized by early involvement of team members, shared risk and reward based on project outcome, joint project management, liability reduction among IPD team members and joint validation of project goals (IPDA 2018). These principles are notably reflected in the CCDC-30 contract, published in 2018 by the Canadian Construction Documents Committee, although there are several standardized models of contracts used in jurisdictions including the United Kingdom and the United States such as the NEC4 Alliance Contract, TAC-1 Term Alliance Contract or the American Institute of Architects series. Regardless of the model, these contracts are based on common principles of common governance, a no-blame culture and the development of a target cost enabling the sharing of profits or losses. IPD is designed to help public bodies achieve functional, environmental, and economic objectives through upstream design iterations involving all relevant stakeholders, decision-making driven by performance objectives, ongoing value management, effective and open communication and the maintenance of quality assurance throughout the process (Jobidon et al. 2019). BIM benefits include faster and more effective processes, better design and production quality, controlled life-cycle costs and automated assembly, while IPD helps achieve better quality levels, shorter completion time, fewer change orders and lower costs (Azhar 2011; El Asmar et al. 2013).

In the province of Quebec, the Société québécoise des infrastructures (SQI) has been tasked with BIM implementation in public projects. The SQI is responsible for managing projects and assets for most of the province's infrastructure projects and serves as a project manager for other public entities. The SQI has implemented BIM in 10 major infrastructure projects so far, with the intention of implementing it in all of its projects by 2021 (Société Québécoise des Infrastructures 2020).

This paper addresses the following research question, or puzzle: How do norms, whether legislative, regulatory or contractual, functionally or dysfunctionally affect the effective implementation of BIM in Quebec's public infrastructure framework? This paper is based on an analysis of relevant literature regarding BIM, IPD, procurement processes and collaborative practices, as well as Quebec's legislation, regulations and contractual documentation regarding the five most recent major infrastructure projects. Four main themes emerged from this analysis and are addressed in this paper using dialectics and the function–dysfunction dyad: collaboration in the tendering process, award criteria, prequalification of tenderers as well as risk and reward sharing. The authors suggest that the use of IPD should help mitigate legal barriers hindering BIM implementation.

BIM and IPD represent a paradigm shift from the traditional, fragmented, linear and adversarial culture of the construction industry to a more trust-based, collaborative and multidisciplinary approach (Lichtig 2006). Although BIM, IPD and collaborative procurement practices are independent of one another, their combination should help public bodies decrease project costs, increase productivity and quality and reduce project delivery time (Azhar 2011). While law has a predominantly territorial nature, the findings

of this paper can apply, with slight variations, to other jurisdictions looking to implement BIM in public infrastructure projects.

## 2. Methodology

The methodology of this paper lies in the development of a question into a research puzzle, which requires asking "what is puzzling about how earlier research has described or explained this (allegedly puzzling) phenomenon?" Essentially, it requires one to ask a "why x despite y" or "how did x become possible despite y" (Gustafsson and Hagström 2018). Applied to the current subject, the puzzle is thus: How can BIM level 3 be implemented despite the normative framework hindering its use? To resolve this research puzzle, this paper relies on di and the use of the function–dysfunction dyad. Although there are many conceptions of dialectics, in each of them intellectual conflicts are developed and resolved, as opposition is their common principle.

In this paper, dialectics reasoning is aimed at overcoming the duality of the function and dysfunction of norms, to achieve a higher order of integration in the form of a synthesis. Quebec's legislation, regulations and contracts are analyzed through hermeneutics, which aims to make sense of an object of study, whether texts or text-analogues. To do so, the contractual documents for Quebec's five latest major infrastructure projects were analyzed. Norms, whether legislative, regulatory or contractual, serve a purpose, or a positive function which acts as one pole of the dialectical spectrum. For example, regulatory norms regarding the award criteria ensure the fair and equal treatment of tenderers. However, they also lead to dysfunctions, the other pole of the dialectical spectrum, such as unduly advantaging price to the detriment of quality. This theoretical framework has notably been used in the study of formal and informal governance mechanisms in public projects, sustainable contracts and statutes analysis (Perillo 1974; Marchais-Roubelat 2012; Howard et al. 2019).

Furthermore, this paper is based on relevant literature regarding BIM, IPD, procurement processes and collaborative practices, which are thoroughly used in the "BIM-specific requirements" subsections of this paper. Finally, this paper represents the third part of a thesis, and thus follows two papers concentrated on a comparative law analysis and a content analysis of different project delivery methods in terms of contractual language (Jobidon et al. 2018, 2019). The results from these papers helped shape the subsections of the current analysis, which address collaborative mechanisms in the tendering process, award criteria and prequalification of firms as well as risk–reward mechanisms.

The following flowchart in Figure 1 illustrates the methodology used in this paper. Each subsection is structured to present the current rules and their functions, BIM-specific requirements, dysfunctions created by the current rules and the tension resolution to achieve successful implementation of BIM.

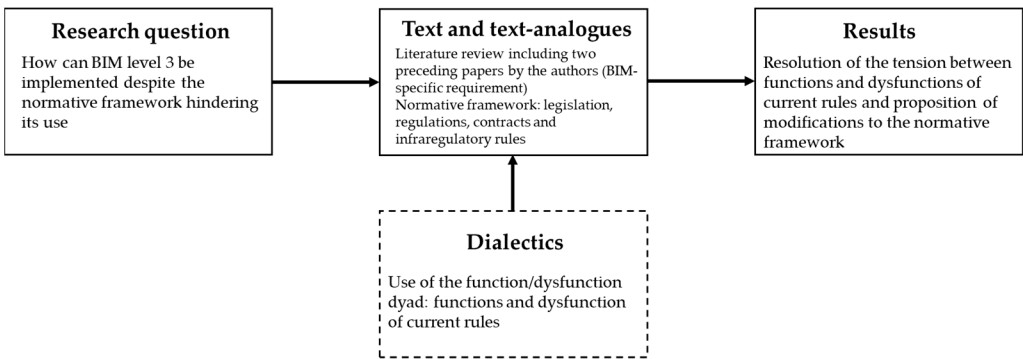

**Figure 1.** Methodology flowchart.

### 3. Facilitating Collaboration and Team Integration

In this section, Quebec's legislative and regulatory instruments are analyzed in terms of collaborative mechanisms, solicitation methods, adjudication criteria and integration of small and medium-sized enterprises (SME), suppliers and manufacturers. We argue that legislation and regulations should officialise and crystallize collaborative procurement practices, enable the evaluation of objective as well as subjective award criteria and standardize the prequalification mechanism.

#### 3.1. Collaboration in the Tendering Process

Quebec's legislation and regulations are mostly silent regarding collaborative practices in the procurement process. Instead, those practices are included in the request for qualifications (RFQ) and the request for proposals (RFP) contractual documents. Three main mechanisms are provided for requests for information (RFI), requests for optimization measures (ROM) and commercially confidential meetings (CCM).

RFIs allow tenderers to clarify, lightly correct or modify project requirements. If the RFI is not confidential in nature, the response is published to all tenderers, thus ensuring the honest and fair treatment of tenderers. If confidential, the response will be communicated only to the appropriate tenderer. ROMs, also subject to confidentiality, aim to significantly alter the technical requirements while ensuring the optimization of quality, costs or delivery schedule of the project. Tenderers can propose solutions that would not be valid without changes to project requirements.

Mandatory CCMs allow tenderers to provide comments and ask questions to facilitate their understanding of the project and ultimately develop compliant proposals. While CCMs have become a staple in the procurement for Quebec's latest major infrastructure projects, public bodies need a derogation from the Treasury to use them since the legislative and regulatory framework do not specifically provide for them. CCMs differ from integrated design workshops—where participants can suggest solutions, freely interact and create value for the project—and rather represent a compliance validation exercise regarding project requirements.

#### 3.1.1. Functions of Current Rules

These three mechanisms ensure a better understanding of technical, functional, commercial and legal requirements, and in the end, the conformity of proposals. They guarantee the fair and equal treatment of tenderers by sharing responses to non-confidential ROMs and RFIs, while enabling innovation through the ROM mechanism. Public bodies have a rather passive role in the process by receiving questions, comments and documents and performing a feedback exercise to validate the compliance of proposals.

#### 3.1.2. BIM-Specific Requirements

Collaboration, coordination and information-sharing are essential to achieve successful implementation of BIM (Antwi-Afari et al. 2018). Project delivery methods used in conjunction with BIM, such as IPD, also necessitate collaboration and effective communication (Sacks et al. 2018). BIM projects can facilitate collaboration, commitment from team members and enhance information-sharing using digital collaboration platforms and workshops (Olatunji 2011).

The SQI's BIM framework, which includes a BIM application guide and a BIM management plan, puts forth collaborative practices and tools to ensure value creation throughout the realization of the project (SOI 2016a, 2016b). These mechanisms include a digital collaboration platform as well as visual coordination, interference detection and integrated design workshops. These practices underpin the quality control process of the concept and serve as communication tools to support decision-making during design development and multidisciplinary workshops (Jobidon et al. 2019).

The BIM application guide also introduces the notion of the master team, also known in other jurisdictions as the planning, design and compliance team, which is responsible

for developing the early design before the selection process. 3D models developed by the master team are part of tendering documents. The design models prepared by the master team must be updated throughout the selection process to include the modifications made by addenda (SOI 2016a).

### 3.1.3. Dysfunctions of Current Rules

The absence of clear directives and guidelines creates a normative fog surrounding collaborative mechanisms and causes uncertainty for public bodies as well as tenderers. Information-sharing and collaborative processes are also hindered by the hyphenation or fragmentation of the process and teams. Finally, rules applicable to fairness interfere with value-creation during the selection process.

### 3.1.4. Tension Resolution

The necessity to obtain the Treasury Board's approval through the derogation procedure creates legal and operational uncertainty for public bodies as well as tenderers since public bodies cannot rely on standardized procedures, guidelines or manuals to adequately supervise the collaborative mechanisms essential to the implementation of BIM or IPD. Furthermore, the RFI, ROM and CCM mechanisms represent a proposal conformity validation exercise and not BIM collaborative workflows such as group modeling or interference workshops. Since there are no mandatory topics in CCMs, there is no obligation, or guidance, to connect the stakeholders' BIM experts to analyze or assess the tenderer's team capacity to interact, cooperate and collaborate in person. Coupled with the legal uncertainty caused by the lack, see the inexistence, of jurisprudence concerning collaborative procurement practices in Quebec, public bodies and tenderers are stuck in a normative fog complexifying the pathway for integrated practices implementation. Infra-regulatory rules such as directives, manuals or guidelines could help clarify the expectations of tenderers, standardize collaborative mechanisms, reduce legal and operational uncertainty and gain predictability for public bodies and tenderers.

The current selection process is also characterized by a hyphenation between four major entities: the client, the SQI, the master team and tenderers. This hyphenation is only filled by the selection of candidates who have formally rather than substantially demonstrated their ability to carry out a project in a BIM context and by the transmission, at the RFP stage, of documentation and data regarding BIM. This hinders upstream contributions to the project and is in contradiction with the Paulson curve, whereby the more changes in a project are made upstream of the process—that is, at the time of design—the less expensive they will be to implement (Paulson 1976). This hyphenation is furthered by the absence of a mention in contractual documents about whether the reference models for the realization phase of the project are the ones developed by the master team or the ones advanced by the tenderers during the selection process. Stating in the contractual documents that 3D models developed by the winning tenderer serve as the basis for future development of the design could effectively ensure continuity and clarity during the selection process.

Another issue is finding balance between sharing non confidential ROMs and RFIs to ensure fairness and the necessary confidentiality to ensure tenderers can add value to projects without losing their competitive advantage. A possible solution would be to consider a more punctual vision of fairness, that is up to a submission of an initial proposal. From that point on, public bodies could entertain bilateral negotiations with one or multiple tenderers, whether through a rank-and-run or best and final offer (BAFO) process to ensure value creation.

Rank-and-run enables public bodies to engage in negotiations with the highest scoring tenderer and with subsequent tenderers in case the initial negotiation fails, while BAFO allows public bodies to entertain individual discussions with each tenderer to enhance their propositions before a final proposal submission. Negotiations and discussions must ensure the fundamentals of the solicitations as well as those of the proposals are preserved and allow the public body to iron out the finer and more confidential aspects of proposi-

tions, whether technical or financial, to achieve more value for the project (Lawther 2007). Including these processes in Quebec's legislation and regulations would help strike balance between fairness and value-creation.

*3.2. Award Criteria*

Quebec's construction contracts regulation provides for either a one-stage lowest tenderer approach or a two-stage RFQ/RFP process for construction works. During the RFQ, the quality of tenderers is evaluated using a minimum of three criteria, which may notably include similar projects recently executed, the experience of a contractor, the ability to ensure efficient project management and the experience of key personnel. The second stage consists of inviting selected contractors to submit a tender including a price and the contract is awarded to the lowest compliant bidder. Quebec's service contracts regulation provides only for quality evaluation of professional service providers through multi-criterion weighting. The contract is awarded to the tenderer obtaining the highest final score.

As for mixed contracts for construction work and professional services, public bodies can use one-stage or two-stage processes. In a one-stage process, public bodies must use multi-criterion weighting and the k coefficient formula. The contract is awarded to the lowest adjusted price tender. In a two-stage tendering process, the RFQ is used to evaluate the quality of tenderers and the RFP can either evaluate only price or price/quality. In the case of the former, the contract is awarded to the lowest compliant tenderer while in the latter, it is awarded according to the lowest adjusted price tender.

### 3.2.1. Functions of Current Rules

The main function of the legislation and regulations is to award public contracts in the most objective way possible to ensure equal treatment of tenderers and sound management of public funds. This is achieved either through multicriteria ponderation of objectivized quality or through the lowest compliant bidder mechanism. Much has been said and written concerning the latter and its limitations, but it should be noted that it is used in a limited way for major infrastructure projects, especially with the rise in popularity of more integrated delivery methods such as design–build (DB).

### 3.2.2. BIM-Specific Requirements

Low-bid, price-driven competition leads to adversarial relationships as well as an increase in costs, schedule delays and poor quality (Lichtig 2006). Procurement models ensuring the integration of team members during the early design stages maximizes the benefits of a BIM project (Porwal and Hewage 2013). Single stage procurement hinders full BIM adoption notably because contractor bids come too late in the process and there is little scope to agree on improvements with the winning team before commencement of construction (Mosey et al. 2016).

Early contractor procurement models include two-stage open book, which invites tenderers to bid for a project based on an outline brief and cost benchmark (Cabinet Office 2014b). The first stage is similar to the RFQ process, but with only one team being selected on their capacity, capability, stability, experience and strength of their supply chain, plus their profit. In the second stage, the chosen team prepares a proposal based on an open book cost which complies with the client's requirements and cost benchmark (Cabinet Office 2014b). Other early involvement procurement models include early BIM partnering and construction management (Porwal and Hewage 2013). These models ensure cost savings, improved design, risk management, sustainable solutions and stakeholder consultation (Mosey et al. 2016).

### 3.2.3. Dysfunctions of Current Rules

Quebec's adjudication criteria still lean heavily on price. As recently as 2018, draft regulations for the procurement of professional service providers prescribed the use of a

price–quality formula unduly advantaging price (AAPPQ 2019). This is the same formula applicable to mixed contracts for professional services and construction works which allows a maximum k coefficient of 15%, thus implying the same overweighting of price. Quebec is the only province in Canada that formally limits the weighting of quality in the award criteria for public infrastructure projects. Furthermore, quality is formally evaluated to ensure objectivity and without evaluating the propensity to collaborate, thus evacuating the essential aspect of human nature in BIM projects, which can lead to conflicts in an environment based on collaborative work and interactions.

### 3.2.4. Tension Resolution

The two poles at play embody an opposition between a purely objective vision of equal treatment of tenderers through price-driven competition, and the need to appreciate parties' behaviour and the inherent qualities necessary to achieve optimal collaboration, quality and value in a BIM context. While some aspects may be objectively quantifiable and qualifiable such as price, experience of tenderers or the number of similar projects, the criteria used to help define quality mostly represent an attempt to objectively assess a subjective matter, a daunting task since quality is easier assessed ex post rather than defined ex ante (Jobidon et al. 2018). These formal criteria give little to no help to public bodies wishing to select a collaborative partner for the realization of a project. The very essence of the BIM collaborative process is the human nature and the participants' interactive qualities, since BIM is considered 10% technology and 90% sociology (Paranandi 2015). The ability to communicate clearly, open-mindedness, walking the extra mile, cooperative behaviour, trustworthiness and creativity are essential qualities, although complex to assess and evaluate.

Once again, a possible solution is the use of a rank-and-run or BAFO process. The RFQ and RFP processes should help public bodies select an appropriate tenderer on an objective quality basis. A more subjective evaluation of tenderers, assessed through predetermined rules in contractual documents, could include interviews with prospective team members and real time sample problems relating to BIM, such as interference detection workshops, for tenderers to demonstrate their ability to work collaboratively (IPDA 2018).

Following this step, public bodies and the top-ranked tenderer could enter a second step during which team alignment and contract negotiation workshops are held to ensure cohesion and the implementation of collaborative practices (IPDA 2018). This stage would also serve to develop a binding target cost. If for some reason previously stated in the RFQ and RFP documents, whether for failing to agree on commercial terms or the tenderer's lack of collaboration, this process was to fail with the top-ranked tenderer, public bodies could go to the next-best ranked tenderer and start all over again. This type of process would help strike balance between the evaluation of objective qualities and the more ineffable ones necessary to achieve fully collaborative BIM, while also ensuring fairness.

Delivery methods other than integrated ones can benefit from BIM. Therefore, it is important to address Quebec's mixed contracts price–quality formula. An evaluation of the price–quality formula using the k coefficient has recently been conducted in Quebec. It was found that in more than 74% of the cases studied, the variation of the k coefficient makes no difference in the choice of the tenderer (AAPPQ 2019). The regulatory requirements regarding the k coefficient are too low for quality to really have an impact. The study suggests that the federal formula, which gives a maximum of 90% of the score to quality and 10% to price, makes quality the paramount adjudication criterion (AAPPQ 2019). When this formula is used, the firm obtaining the best quality rating is favored in all cases and configurations of procurement compared to the firm offering the lowest price. Quebec should therefore consider incorporating the federal formula, or a version thereof, to ensure that tenderers are selected based on the quality of their propositions and not only, or mostly, their price.

*3.3. Prequalification to Ensure Integration*

One important legislative principle is the opportunity for qualified tenderers to compete in calls for tenders, which includes SMEs, suppliers and manufacturers. Since 2016, public bodies must adopt guidelines which must ensure openness of public markets to competition and SMEs (Conseil du Trésor 2019). This has led to various practices such as inviting at least one SME in an invitational tender, and the creation of a public market accessibility index, which denotes a high degree of variation in terms of best practices for SME inclusion (Conseil du Trésor 2019). Quebec also offers financial support to companies wishing to acquire the required equipment and software to use BIM (Gouvernement du Québec 2016). Furthermore, Quebec's regulations provide for prequalification of contractors and service providers.

DB contracts in Quebec include provisions mandating the creation of a project management control committee, composed of public and private actors, whose role is notably to review all matters concerning design and construction issues. The committee, at its discretion, may invite any relevant party to meetings, which broadens the possibility of key stakeholders' involvement. The design builder can also voluntarily implement multidisciplinary workshops. Furthermore, BIM projects are usually coupled with publicly mandated integrated design workshops, during which any collaborator can be invited, such as a manufacturers or end-users (Jobidon et al. 2019).

### 3.3.1. Functions of Current Rules

Rules and regulations aim to entertain healthy competition by ensuring the fair treatment of tenderers and by giving qualified tenderers and opportunity to compete. Regulations enable the use of a flexible prequalification mechanism, while some contracts allow the inclusion of key stakeholders during various project stages. Although silent in regard to SME or manufacturer inclusion, legislation, regulations and contracts are complemented by different governmental initiatives.

### 3.3.2. BIM-Specific Requirements

Since BIM intends to support a more integrated team approach, procurement models need to emphasize the early contributions of contractors and specialist contractors to the BIM model to develop functional specifications and thus facilitate information management, communication and collaboration (Vidalakis et al. 2020).

SMEs barriers to BIM implementation notably include legal ambiguity in terms of roles, responsibilities and distribution of benefits (Sun et al. 2017). The prefabrication industry, for which BIM can serve as a helpful tool to facilitate on-site assembly services, can benefit from the BIM collaborative environment and visualization of the physical and functional representations of prefabricated components (Khosrowshahi and Arayici 2012). However, few prefabrication projects have benefited from BIM, and vice versa, notably because of a lack of common BIM standards and of understanding firms' readiness to adopt BIM (Khosrowshahi and Arayici 2012).

Many different frameworks and tools exist to assess firms' BIM performance, such as Succar's BIM capability framework and BIM Quickscan (Mahamadu et al. 2017). Prequalification helps minimize the risk of selecting unsuitable firms for BIM projects by shortlisting potential suppliers and partnering (Porwal and Hewage 2013). Most of the BIM performance assessment models used for prequalification and qualification of suppliers focus on the physical resources and processes required instead of softer measures such as behavioural and organisational factors (Mahamadu et al. 2017).

### 3.3.3. Dysfunctions of Current Rules

Among dysfunctions figure the difficulty to include specialist contractors and manufacturers during the early design stage. Although DB or other early contractor involvement delivery methods allow for the inclusion of different stakeholders, this can only occur once the contract is awarded which means a significant portion of the design has already

been advanced by the master team. The absence of subcontractors and manufacturers in the selection process can also lead to inequities regarding risk–reward sharing and hinder information management and the harmonization of BIM processes. As of right now, no prequalification for SMEs, specialist contractors or manufacturers has been published in Quebec's electronic tendering system. Quebec has also not provided for guidelines, manuals or any form of standardization of BIM capability for prequalification.

### 3.3.4. Tension Resolution

BIM projects necessitate the inclusion of SMEs, manufacturers and specialist contractors in early design stages, but not at the expense of fairness. The only way to assess firms' skills, capacities and maturity regarding BIM, such as prefabrication companies, remains the RFQ process, which requires time and money both for public bodies and tenderers, and which is based on non-standardized quality criteria rather than BIM-specific indicators.

Including a standardized questionnaire, such as the UK's PAS 91:2013, in the prequalification mechanism would reduce legal uncertainties and allow the evaluation of BIM capability, notably the interoperability of software and models, process harmonization and staff training, while still preserving fairness (British Standards Institution 2013). A standardized questionnaire, which focuses on physical resources and processes, should be complemented with the evaluation of softer factors such as behavioural attributes and collaborative attitudes, since these factors influence BIM delivery success (Mahamadu et al. 2017). Interviews with prospective team members and sample problems relating to BIM would serve this purpose. The questionnaire could also give the public sector quality information on the capacity and maturity of the market, and firms could adjust their practices to those implemented in public projects.

The use of IPD would allow the integration of specialized contractors and manufacturers upstream of the project while including them in team alignment and contract negotiation workshops (IPDA 2018). It is necessary to include the right people in the project team before their tasks are to be performed, hence the importance of the presence of specialist contractors and manufacturers early in the process to notably help with the development of the target cost (Zimina et al. 2012).

## 4. Enabling Risk and Reward Sharing

This section addresses Quebec's rules applicable to risk–reward allocation mechanisms, notably through remuneration regulations, intellectual property, stipends, liability and insurance. We argue that legislation, regulations and contracts should move from risk–reward allocation to a risk–reward sharing paradigm.

### 4.1. Sharing Risks

The construction contracts regulation provides compensation for tenderers when the selection process is canceled. Tenderers' compensation is CAD 5000 for projects with a value greater than CAD 1 million. Quebec's latest alternative delivery methods RFQ and RFP documents diverge from this regulatory standpoint by offering significantly larger stipends to unsuccessful but compliant tenderers, thus indicating the use of the derogation procedure.

In traditional delivery methods, the public sector is notably responsible for planning, design, operation, maintenance, financial and legal risks, while the private contractor is mainly responsible for execution of the works. Alternative delivery methods transfer a larger share of the risk to the private sector. Subject to the public body's risk transferring decisions as negotiated in the selection process, Quebec's contractual documents state that contractors are notably responsible for permits and authorizations, design, construction, respect of costs and schedule, insurance and project management.

Insurances include a construction all risks insurance, which provides protection against loss or damage regarding works, equipment and machinery and third-party claims for property damage or bodily injury. Contractors in DB projects must also take a wrap-up

liability policy as well as professional liability to cover losses resulting from any error or omission in design and construction. Contractors also need performance bonds to cover their contractual obligations, as well as a labor and material payment bond to ensure the payment of subcontractors and suppliers for their work and the material they supply. Finally, a parent company guarantee might be necessary to protect the client in the event of a contractor's default.

### 4.1.1. Functions of Current Rules

Regulatory stipends ensure light compensation for firms participating in a canceled selection process while stipends provided through the derogation procedure aim to share the financial risks associated with propositions development. Public bodies aim for the optimal risk transfer to the private sector by pegging risks to the party best positioned to manage it. Insurance, to protect the insured against claims, and liabilities, to protect losses incurred by third parties, enable risk distribution and contingency provisions. Performance bonds guarantee against the failure of the other party to meet contractual obligations and ensure claims against the other party in the case of default.

### 4.1.2. BIM-Specific Requirements

Tenderers deliver a substantial design effort in preparing proposals, especially in BIM projects which necessitate extra efforts in the early design stage, and stipends compensate this effort. Stipends increase competition, bidding pool diversity, SME inclusion in public procurement and quality of proposals, while the design level of effort reduces cost growth, and their absence can lead contractors to not participate in the process. The appropriate level of a stipend is somewhat flexible, but the rule of thumb for a two-stage process is one-third of the design effort (Alleman et al. 2020).

Level 3 BIM, through integration of the team, may blur the levels of responsibility and enhance risk and liability (Azhar 2011). Risks must be identified and allocated, especially regarding responsibility for the accuracy and coordination of data as well as updating information in collaborative models (Porwal and Hewage 2013). The BIM addendum addresses the risk of project participants assuming contributions made by other parties are accurate and provides for claim waivers (Porwal and Hewage 2013). The BIM addendum also specifies that the participation of the contractor, subcontractors and suppliers in a model does not constitute design services, although this cannot be applied to DB because of its single point of responsibility, which means parties only assume their traditional roles (Currie 2014). CIC BIM Protocol limits project team members' liability by stating that there is no warranty to the integrity of electronic data transmission and no liability for corruption or alteration occurring after transmission (Mosey et al. 2016).

Traditional insurance is not easily adaptable to BIM level 3, and new insurance products better tailored to collaborative projects are needed (Currie 2014). One possible solution is Integrated Project Insurance (IPI), which can be used in alliancing or IPD models. This model covers all major parties under one single policy and includes all the insurances needed for infrastructure projects (Currie 2014).

### 4.1.3. Dysfunctions of Current Rules

Regulatory stipends do not reflect the actual work carried out by tenderers in BIM projects, and public bodies must use the derogation procedure to offer better compensation. Absence of stipends can disinterest firms and competition and favor firms with stronger financial records to the detriment of smaller ones. Risks are allocated and not shared, which does not reflect the collaborative BIM process and furthers the silo effect. Furthermore, Quebec's contractual documents do not provide for BIM-specific liability limitations or claim waivers.

#### 4.1.4. Tension Resolution

Stipends provided for through the derogation procedure better compensate tenderers for their proposal development efforts. Since tenderers' intellectual property developed throughout the selection process is transferred to public bodies, stipends merely represent small compensation for the accomplished work. Stipends do not need to cover the full costs of preparing the proposal, as there is an inherent cost to business development which should not be fully borne by the state. Several jurisdictions reasonably compensate tenderers to ensure healthy competition and involve smaller players in the selection process. Quebec's regulations must be revised to benefit from stipend's advantages and lessen public bodies' administrative burden regarding the derogation procedure.

IPD is one of the most effective ways of dealing with BIM-specific risks because of its pain–gain sharing mechanism (Azhar 2011). The target cost includes a single project contingency intended to reduce construction costs. The cost-sharing mechanism must define the risk–reward proportions of parties whose overhead costs are at-risk (Zhang and Li 2014). IPD also waives or limits claims against the other parties (Ashcraft 2008). Liability limits such as the ones provided for in the CIC BIM Protocol help strive towards a no blame culture (Currie 2014; Mosey et al. 2016). The CCDC 30 contract stipulates that parties must waive all claims against each other.

IPI has been proposed as a solution to the issue of the blurred levels of responsibility in BIM level 3, and aims to align parties' interests, ensure the development of achievable and affordable solutions and cover project outcomes rather than individual liabilities, such as insuring the potential cost overrun (Currie 2014). IPI collectively insures all partners of an alliance or IPD model, such as the client, designers, consultants, manufacturers, constructors and their supply chain. In IPI, insurers are more involved in the project through the participation of an independent facilitator and a technical independent risk assuror (Cabinet Office 2014a; Currie 2014). With IPI, parties' contributions are not fixed and can be reallocated during the project. Disputes could thus arise amongst project participants, without affecting the client, but it also could encourage the exercise of reasonable skill and care (Currie 2014).

In order to move from a risk allocation to a risk-sharing paradigm, Quebec should therefore consider revising the regulatory indemnity regime, use the IPD pain–gain sharing mechanism of target costing and include waivers and liability limitations in its contractual documents, as well as enter discussions and negotiations with the insurance industry to develop an IPI model to move towards a no-blame culture.

#### 4.2. Sharing Benefits

BIM-specific services are considered as additional, or special, services under the regulatory fixed-remuneration scales for professional services (AAPPQ 2016). Those services are not included in the applicable percentages for services rendered during preparation of plans and specifications, whether preliminary or final, and during construction. Public bodies pay for these services on a lump sum basis, provided the scope is well defined, negotiated on the basis of an estimate of the number of hours necessary to complete the services (AAPPQ 2016). As for engineers, their professional order does not offer any guidance on the topic.

Canada's Copyright Act provides that architects can claim copyright ownership on drawings, a fixation of an original idea, a principle also present in the Fee Rate for Professional Services Provided to the Government by Architects (FRA) and the Fee Rate for Professional Services Provided to the Government by Engineers (FRE). It also states that joint authorship is possible in situations of collaborative or collective works. Quebec's contractual documents provide that exclusive intellectual property is transferred from contractors to public bodies, which in exchange grant contractors a license. Project data are transferred to public bodies for the management and operation of the building, as well as for future projects. These data, including all copyrights attached, become the exclusive property of public bodies (AOI 2016b).

### 4.2.1. Functions of Current Rules

Remuneration regulations value fairness through standardization, which reduces uncertainty and enables budget control. Contractual payment mechanisms in traditional delivery methods reduce financial pressure, while milestones transfer the risks of temporary financing to firms and ensure completion of projects. The legislative intellectual property regime creates economic rights with the purpose of providing payment to the author or copyright owner. Legislation provides for joint authorship, but not contractual documents which instead ensure acquisition of intellectual property on a fixed price basis for potential future works.

### 4.2.2. BIM-Specific Requirements

Since BIM improves service delivery, there is a need for commensurate compensation through scales of fees (Hamil 2012). Change of standard in professional practice, notably through clash detection, development of 3D models, simulations, training, time inputting, reviewing and transferring usable data to public bodies, should lead to a design fee rise for designers (Ashcraft 2008). While traditional delivery methods hinder performance-based remuneration because of silos, alternative delivery methods have been found to better distribute the benefits (Sacks et al. 2018).

Issues of model ownership need to be stipulated and standardized to facilitate BIM implementation. Ordinarily, the ownership of the design belongs to the designer following the completion of a project, but since BIM facilitates infrastructure management, models have a significant value for public bodies, which should use and develop BIM in the entire project life-cycle (Porwal and Hewage 2013). Most BIM manuals state that public bodies are the owners of the digital models, information and other deliverables (Sacks et al. 2018).

### 4.2.3. Dysfunctions of Current Rules

The FRA and the FRE, the remuneration regulations applicable to professional services, have not been substantially revised since 1984 and do not reflect the computerization of professional practice, while fee scales have not been indexed for 9 years. Contractual payment schemes are task-led instead of performance-led, and progressive payments do not reflect the additional efforts needed in the early design stage of BIM projects. The rules applicable to benefits harm team integration, further the silo effect and do not reflect the collaborative and multidisciplinary reality of BIM (Ghassemi and Becerik-Gerber 2011). Quebec's contracts do not address the issue of joint authorship when BIM level 3 sometimes makes it impossible to determine where the contribution of one party ends and the other begins.

### 4.2.4. Tension Resolution

Balance must be struck between the desire to treat all firms equally through anticipation of the terms of the exchange, a form of fairness, with providing the necessary structure to facilitate collaborative processes in BIM projects. It is necessary to undertake a revision of remuneration regulations through negotiations between the state and professional associations to mutually and adequately adapt them to the computerization of professional practice. This adaptation should consider the efforts provided to generate quality information to meet public bodies' needs. This mutual revision would ensure fairness, whether through the negotiation process or regarding the final fee scales.

Repartition of payments should roughly be spread out evenly during the four major stages since the early design stage of BIM projects is more intense. To ensure collaborative work and a focus on project goals, BIM projects must shift from task-led to performance-led payments. The IPD compensation mechanism ensures participant's success is tied to the overall project success (O'Connor 2009). This compensation mechanism relies on the implementation of a target cost combined with an estimated maximum price during the negotiated phase of the RFP, which enhances value for budgeting (Chan et al. 2011). The estimated maximum price can motivate the project team to achieve better value by aligning

their financial objectives with that of the project (Darrington and Lichtig 2010). Corollary, the tenderers' ability to develop a target cost should be thoroughly evaluated during the first stage of the RFP.

The target cost is comprised of reimbursable costs, which are not at risk and include direct and indirect costs such as overhead costs specific to the project, project-specific costs and risk contingencies, while profit margins and company overheads are at risk (O'Connor 2009). Savings on the actual costs, as compared to the target cost, can be shared according to agreed-upon percentages, although public bodies could opt to set tentative percentages with arbitral adjustments to avoid gross inequities that could result from the set percentages (O'Connor 2009).

As for intellectual property, it has been found that contractual documents that not providing for joint authorship could discourage collaboration at an advanced level, especially with BIM level 3 (Currie 2014). Quebec's contractual documents should include the concept of joint authorship by defining it and by recognizing the right of the original author to accept or reject additions. The original authors would thus be saved from any liability if an erroneous addition is made without their consent (NBIMS 2007).

## 5. Conclusions

This paper notably highlights the need to rely more heavily on legislation, regulations, contracts and infra-regulatory rules to clarify public bodies' expectations, to standardize collaborative mechanisms, to reduce uncertainty and to clarify the status of the model between the procurement and the realization phase. Contractual mechanisms such as requests for information, requests for optimization measures and commercially confidential meetings provide some form of collaboration, but none of these mechanisms allow the assessment of tenderer's BIM competence before the adjudication of the contract. Tenderers are thus not evaluated, or very lightly so, regarding their capacity to carry out BIM projects and when they are, it is formally rather than substantially, such as through interactive scenarios or problem solving. This is partly caused by the imperatives of fairness whereby public bodies must publicly share responses to requests for information and requests for optimization measures. One possible solution to this problem would consist of integrating solicitation methods, such as rank-and-run or best and final offer (BAFO), which would enable public bodies to enter bilateral negotiations with the highest scoring integrated team of the first stage of the request for proposals, thus ensuring the refinement of more confidential aspects of the proposals, whether technical or commercial, during the second phase of the request for proposals.

Since BIM is 10% technology and 90% sociology, it is essential to adjust award criteria for integrated projects. The tension between the desire to ensure equal treatment of tenderers objectively and formally through price-based competition and the need to assess the inherent qualities of the parties in order to achieve optimal collaboration must thus be resolved. The essence of BIM lies in human interactions, clear communication, open-mindedness and cooperative behaviour. The use of negotiated two-stage procurement processes can help resolve this tension. The request for qualifications and the first stage of the request for proposals process would allow public bodies to assess objective qualities, such as experience and similar projects carried out, and subjective qualities through interviews and real-time problem-solving scenarios. The second stage of the request for proposals would ensure the co-development of technical and financial matters through multiple workshops enabling the development of trust and the harmonization of processes between public and private parties.

Furthermore, the use of a negotiated two-stage procurement process—during which a risk–reward sharing mechanism is developed through target costing combined with an estimated price—as well as incorporating an integrated project insurance model, claim waivers and liability limitations should allow better team integration and collaboration. Additionally, since BIM and IPD encourage upstream participation from a wide range of stakeholders, suppliers and manufacturers could be prequalified by completing a standard-

ized questionnaire to assess their BIM capability and maturity, while also assessing softer traits such as behavioural attributes and the propensity to collaborate. Their presence and participation in the request for proposals process would facilitate target costing and the inclusion of innovative delivery solutions for the project.

The importance of preliminary design in IPD and BIM shifts design efforts to the early stages of the project. However, regulatory remunerations have not been updated or indexed for 9 years to reflect the evolution of professionals' practice resulting from new technologies and the early-stage intensity of design. The benefits do not only take pecuniary forms but are also reflected in intellectual property. Contractual documents should therefore reflect the interwoven, multidisciplinary and collaborative design processes that blur the lines of parties' roles and responsibilities and thus provide for joint authorship.

The authors suggest that the use of IPD should help mitigate legal barriers hindering BIM implementation, while preserving balance between fairness and encouraging collaboration. This would ensure public bodies reap the full benefits of the BIM process such as better design, controlled life-cycle costs, production quality, automated assembly, cost savings and reduction in project time. To do so, revisions must be made to legislative, regulatory and contractual norms to facilitate an optimal implementation of BIM for future public infrastructure projects to enable the achievement of the public interest by obtaining.

**Author Contributions:** Conceptualization, G.J.; methodology, G.J.; validation, G.J., P.L. and R.B.; formal analysis, G.J.; writing—original draft preparation, G.J.; writing—review and editing, G.J., P.L. and R.B.; supervision, P.L. and R.B. All authors have read and agreed to the published version of the manuscript.

**Funding:** The authors are grateful to the Natural Sciences and Engineering Research Council of Canada for the financial support through its IRC (IRCPJ 461745-12) and CRD (RDCPJ 445200-12 and RDCPJ 445200-12) programs, as well as the industrial partners of the NSERC industrial chair on eco-responsible wood construction (CIRCERB).

**Institutional Review Board Statement:** Not applicable.

**Informed Consent Statement:** Not applicable.

**Data Availability Statement:** Not applicable.

**Conflicts of Interest:** G.J. is currently employed by the SQI, which had no say in the design of the study; in the collection, analyses or interpretation of data; in the writing of the manuscript, or in the decision to publish the results.

## Abbreviations

| | |
|---|---|
| BAFO | Best and final offer |
| BIM | Building information modeling |
| CCM | Commercially confidential meetings (CCM) |
| DB | Design-build |
| FRA | Fee Rate for Professional Services Provided to the Government by Architects |
| FRE | Fee Rate for Professional Services Provided to the Government by Engineers |
| IPD | Integrated Project Delivery |
| IPI | Integrated Project Insurance |
| RAIC | Royal Architecture Institute of Canada |
| RFI | Requests for information |
| RFQ | Request for qualifications |
| RFP | Request for proposals |
| ROM | Requests for optimization measures |
| SME | Small and medium-sized enterprises |

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
