# Peer review of "Building Information Modeling in Quebec’s Procurement for Public Infrastructure: A Case for Integrated Project Delivery"

_laws, 2021_

Round 1

Reviewer 1 Report

The starting point for this paper is BIM and IPD. What is meant by both of these terms could be more comprehensively explained. IPD should also be defined in terms of the benefits it purports to deliver. The term level 3 is also used this also could benefit with more definition. I also believe some discussion of the IFOA contract would be helpful.

Author Response

IPD: Lines 60 to 70 were added to further explain IFOA agreements and IPD. As for benefits, they were already mentioned in lines 72-73.

BIM: Further explanations on BIM were included. BIM level 3 was further explained in the introduction, notably through a comparison of other BIM maturity levels.

Reviewer 2 Report

Manuscript ID laws-1218324

Title

A dialectical perspective on Building Information Modeling implementation in Quebec’s procurement for public infrastructure: a case for Integrated Project Delivery

This paper explores the possibilities of integrating BIM into public infrastructure contracts. This offers new possibilities for tenderers and would clearly improve the public procurement process, which needs to be as transparent as possible so that competition between tenderers can offer the administration a better service at the best possible price, without the latter being the decisive factor. The manuscript is well written, but I suggest some changes to improve it in case the editor or the authors would like to take them into account.

The title of the manuscript seems too long to me. Although there is no character limit in this journal, I suggest shortening it to make the manuscript more attractive. This is a recommendation, and only if the authors want to adopt the suggestion: “Building Information Modeling in Quebec’s procurement for public infrastructure: a case for Integrated Project Delivery”

The abstract is too short and does not reflect the content of the manuscript. It should be rewritten.

The key words. Some are unnecessary and some are repeated, I propose to select only one of them: Building information modeling or BIM; integrated project delivery or IPD. Should be avoided: construction, contracts

The introduction should be updated with more recent manuscripts on BIM. e.g. (2019). Bibliometric maps of BIM and BIM in universities: A comparative analysis. Sustainability11(16), 4398.

Abbreviations should be cited the first time they are used. E.g.  IPD, FRA, FRE,..

The manuscript could add a table of abbreviations to make it easier for the reader to understand the text.

The methodology section largely reflects the objectives. This part should go in the introduction. I propose that the authors make a flowchart of the methodology used to clarify this for the reader.

The methodology section should not reflect or comment on the results obtained. Line 92 “The results from these papers helped…”

Line 252 “…allows a maximum « k coefficient » of 15%, thus implying the same overweighting of price”. Is this % the same for the tenderers in others place of Canada or only for Quebec?. In other countries there are not this limitation, others are 20 %. Please clarify if it is possible.

The conclusions are appropriate to the manuscript, but I suggest that they should be written without abbreviations and, if possible, without references. Since this section of conclusions closes with a reference, it would be better to close with a conclusion drawn from the manuscript itself.

Some of these conclusions should be reflected in the abstract.

Author Response

Thank you for your review. You will find the point by point response attached.

Round 2

Reviewer 2 Report

The authors have improved their manuscript in response to my suggestions. As far as I am concerned, everything is now correct. The article can be published if the editors of the journal so decide.